# Cancer Loyalty Card Study (CLOCS): feasibility outcomes for an observational case–control study focusing on the patient interval in ovarian cancer

Hannah R Brewer ![ORCID] ,[1] Marc Chadeau-Hyam,[2,3] Eric Johnson,[4] Sudha Sundar,[5] James Flanagan ![ORCID] ,[1] Yasemin Hirst ![ORCID] [6]

For numbered affiliations see end of article.

**Correspondence to**
Dr James Flanagan;
j.flanagan@imperial.ac.uk

## ABSTRACT

**Objectives** Ovarian cancer symptoms are often non-specific and can be normalised before patients seek medical help. The Cancer Loyalty Card Study investigated self-management behaviours of patients with ovarian cancer prior to their diagnosis using loyalty card data collected by two UK-based high street retailers. Here, we discuss the feasibility outcomes for this novel research.

**Design** Observational case–control study.

**Setting** Control participants were invited to the study using social media and other sources from the general public. Once consented, control participants were required to submit proof of identification (ID) for their loyalty card data to be shared. Cases were identified using unique National Health Service (NHS) numbers (a proxy for ID) and were recruited through 12 NHS tertiary care clinics.

**Participants** Women in the UK, 18 years or older, with at least one of the participating high street retailers' loyalty cards. Those with an ovarian cancer diagnosis within 2 years of recruitment were considered cases, and those without an ovarian cancer diagnosis were considered controls.

**Primary outcome measures** Recruitment rates, demographics of participants and identification of any barriers to recruitment.

**Results** In total, 182 cases and 427 controls were recruited with significant differences by age, number of people in participants' households and the geographical region in the UK. However, only 37% (n=160/427) of control participants provided sufficient ID details and 81% (n=130/160) matched retailers' records. The majority of the participants provided complete responses to the 24-Item Ovarian Risk Questionnaire.

**Conclusions** Our findings show that recruitment to a study aiming to understand self-care behaviours using loyalty card data is challenging but feasible. The general public were willing to share their data for health research. Barriers in data sharing mechanisms need to be addressed to maximise participant retention.

**Trial registration number** ISRCTN14897082, CPMS 43323, NCT03994653.

## STRENGTHS AND LIMITATIONS OF THIS STUDY

⇒ This study is the first of its kind to investigate transactional data for cancer outcomes.

⇒ It has established data governance structures and methods to comply with General Data Protection Regulation and privacy policies for using retailer transactional data.

⇒ This study provides a pre-existing prospectively collected data set free from recall biases which is a common bias in case–control studies.

⇒ Recruitment happened during the COVID-19 pandemic, which affected recruitment rates, particularly for cases who were recruited through National Health Service clinics.

⇒ Increasing age was negatively associated with participation rates, particularly in the control population, requiring targeted recruitment to older participants.

## INTRODUCTION

Approximately 7500 new cases of ovarian cancer are diagnosed in the UK each year, and it is estimated that ovarian cancer incidence rates will rise by 15% in the UK between 2014 and 2035.[1] Unfortunately, 23% of patients with ovarian cancer in 2018 were diagnosed at Stage IV[2] when treatment options are less than optimal for survival. Late-stage ovarian cancer diagnosis is often associated with delayed patient presentation and a longer diagnostic interval due to non-specific cancer symptoms (eg, feeling bloated, indigestion, feeling full and abdominal pain). While there has been progress in efforts to reduce diagnostic delays using routine blood tests (ie, CA125)[3–5] there is little evidence focusing on women's self-care behaviours associated with managing non-specific symptoms. It has been demonstrated that population screening of asymptomatic women is not effective in reducing mortality from the disease and,

therefore, is not recommended in the UK.[6] Screening in the symptomatic population may prove more effective,[7] however, the main barrier to earlier diagnosis remains a lack of awareness of the vague non-specific symptoms of ovarian cancer.[8]

The Cancer Loyalty Card Study (CLOCS) investigated the period between the presentation of the first symptom(s)/sign(s) leading up to the first presentation at healthcare providers (ie, the patient interval) of patients with ovarian cancer to better understand self-care behaviours using past purchase information collected through loyalty cards at two high street retailers in the UK.[9] This project aimed to better understand whether self-care behaviours can be detected at an early stage and facilitate earlier presentation to healthcare providers, and the main results have been reported elsewhere.[10] We have previously reported in a proof-of-concept study that loyalty card data may be used to identify an increase in painkiller and indigestion medication purchases among patients with ovarian cancer.[11] The proof-of-concept study warranted testing this hypothesis in a large-scale case–control study design aiming to include 500 patients with ovarian cancer (cases) and 500 individuals without an ovarian cancer diagnosis (controls).[9]

Flanagan *et al* demonstrated the key information governance and ethical considerations of recruiting participants to a project that is requesting consent for transactional data to be used in health research.[11] These included separate consent forms for the study and data requests for each retailer, clear communication with potential participants about which information is being collected, stored, how it is being used, how long and data breach risks, if any. Loyalty card ownership was also identified as a logistical barrier. Thus, ownership was recommended as a prerequisite of participation in the study to avoid data loss or invalidated consents. In terms of data reliability among cases, it was also recommended that patients should be recruited to the study close to their initial diagnosis for researchers to have a retrospective purchase history capturing self-care behaviours before ovarian cancer diagnosis.

Furthermore, recruitment and participant retention to case–control studies are not without challenges; they have often been criticised for potential selection bias and representativeness of control participants against case participants.[12] Traditional recruitment strategies often involve recruitment through healthcare organisations, random digit dialling and postal invitations, recruitment via family and friends of patients.[13] In CLOCS, due to low prevalence of ovarian cancer in the UK, a multisite recruitment approach for cases in the UK was proposed with social media recruitment of controls to minimise selection bias and maximise representativeness.[9] However, while patients' willingness to take part may be higher for cancer research, it was also deemed necessary to improve public acceptability among potential control participants by raising awareness and improving understanding using an accessible communication format, that is, a tailored website (www.clocsproject.org.uk).

The purpose of this paper is to present the feasibility outcomes of CLOCS which aimed to recruit 1000 women with and without ovarian cancer in the UK, describe feasibility of a large-scale case–control study by assessing participant rates, characteristics and any possible changes to the study design that could optimise data collection in the future. In particular, we discuss the barriers and facilitators of recruiting control participants to a case–control study through social media platforms and case participants through ovarian cancer clinics in National Health Service (NHS) trusts in light of the COVID-19 pandemic.

## METHODS

### Design

The aims, design and results of CLOCS have previously been reported.[9 10] CLOCS is an observational case–control study where participants are women in the UK, aged 18 years or older, who hold a loyalty card at either or both participating high street retailers (HSR1 and HSR2). Participants with a diagnosis of ovarian cancer were considered cases, and those without a previous ovarian cancer diagnosis were considered controls.

### Settings

Cases were recruited from 12 NHS clinic sites in England, Wales and Scotland. Eligible patients were identified by each site's recruitment team and presented with information in person about the study in paper format. If they decided to take part, participants were given the consent form, a risk factor questionnaire and a clinical questionnaire (completed by their clinical care team) to either complete in the clinic or take with them to complete at another time (online supplemental material 1). Completed forms were then posted to the CLOCS main research site.

In response to the COVID-19 pandemic, an amendment was submitted and approved by the NHS Research Ethics Committee (20 September 2020, 19/NW.0427-SA1) to allow clinics to recruit eligible patients with ovarian cancer over the phone, post them the study documents to complete where they were shielding and post completed forms to the main research site.

Controls were recruited primarily using paid/non-paid web-based methods, snowball sampling, a pilot-targeted email campaign from one of the participating high street retailers to its users and word of mouth. These methods included paid Facebook advertising, distribution of study blogs on the study website, study twitter (@CLOCS_Imperial), VOICE Imperial College London[14] and widely distributed webinars that aimed to address barriers in CLOCS participation. In October 2021, a pilot email was sent to 1053 cardholders from one of the retailers to test the impact of sending invitations for CLOCS and measure the response rate to invitations.

## Procedures

### Consent+questionnaire completion (step 1)

Each potential participant was provided with a detailed information sheet and privacy policy to facilitate informed decision making. This was followed by study consent, the study questionnaire and a clinical questionnaire for cases. The CLOCS participant consent form consisted of two sections. In the first section, participants consented to take part in the study which provided information on data retention, how data is being handled for the study, how long and how to withdraw. In the second section, participants provided consent for the research team to request their purchase history from the participating retailer(s) using the exact numbers and names on their loyalty cards. Contact details were collected to clarify loyalty card details in the event their details did not match the retailers' records. Participants were given the option to consent to be re-contacted to take part in future related studies.

Case participants completed paper-based forms which were then posted back to the research team for secure data entry. Control participants were asked to complete a web-based secure study form including the information sheet, the consent form and risk factor questionnaire on the study's website (www.clocsproject.org.uk).

### Substantial amendment—ID verification (step 2)

In September 2020, the CLOCS research team received approval from the North West—Greater Manchester South Research Ethics Committee (dated 20 September 2020, 19/NW/0427-SA1) to request identity verification details including a visual proof of ID (eg, passport) and a proof of address (eg, utility bill) from participants. This was a development during the research recruitment process before data sharing agreements with HSR1 and HSR2 were established. Thus, the aim of the ID verification step was to ensure that those participating in the study were not completing the forms on behalf of someone else and could be verified against the high street retailer's records. Therefore, control participants who took part in the study online were asked to provide a copy of a photo ID and proof of address, using a unique and secure weblink, after step 1 was completed. If they did not complete step 2 at the time of the participation, they were sent a maximum of three reminder emails, which included their unique upload link. All ID verification documents were encrypted using private and public keys and kept until data requests from the retailers were completed or when the study was complete, whichever came first. Control participants who took part prior to September 2020 were sent unique secure links to consider uploading this information so that the research team could request their purchase history data from the retailers. If they did not upload the documents after their reminders, no requests were made from the retailers on their behalf. Identification of cases in NHS clinics using unique NHS numbers was sufficient for ID verification. Details of the transfer of participant purchase history between the research team and retailers are described in the protocol.[9]

Participants who completed the consent form and risk factor questionnaire were eligible for the ovarian cancer risk analysis, regardless of whether they provided ID verification or their loyalty card details matched retailer records. We recontacted 39 participants to verify loyalty card details.

### Recruitment status

Recruitment was opened for ovarian cancer patients from 1 November 2019 and for control participants from 1 July 2020. Recruitment was suspended from 31 January 2022 (the original planned closing date) pending funding decisions and was finally closed on 28 July 2022.

### Patient and public involvement (PPI)

PPI informed the design and the recruitment strategies throughout CLOCS. We presented two public facing seminars in December 2020 aiming to engage with the wider audience about CLOCS and changes in the participation criteria using ID verification and in March 2021 during Ovarian Cancer Awareness month aiming to discuss the potential impact for patients with ovarian cancer. Furthermore, we held three scientific meetings annually inviting academics, patient representatives and participating high street retailers.

### Feasibility outcomes measures

The following measures were collected and used to assess feasibility outcomes of CLOCS based on guidance on reporting feasibility outcomes and the reporting of observational studies in epidemiology.

Participant descriptive characteristics: Date of birth was collected and re-categorised in 10-year age groups for reporting. Other characteristics included ethnicity, the number of people in the participant's household (1 to ≥5 persons), self-reported loyalty card usage (not at all/not very often/sometimes/often/all the time/missing), loyalty card membership for the retailers (HSR1, HSR2, both and neither) included in the study and family history of ovarian and/or breast cancer (Yes/No/Missing). We included 17 different ethnic backgrounds to be recorded, due to the small number participants in sub-populations, we categorise this as white/non-white/prefer not to say and missing.

The questionnaire was amended to include a question about whether participants had been diagnosed with COVID-19 (responses limited to the following: diagnosed and recovered, diagnosed and still ill, suspected but not formally diagnosed/did not have COVID-19) at the time of the study participation.

Further data included the Index of Multiple Deprivation (IMD) quintiles based on the postcodes of the stores where purchases were made ranging from most deprived (1) to least deprived (5). The location where the purchases were made was included based on the nine UK regions (East of England & Yorkshire and Humber, London,

North East, North West, Midlands, Scotland, South East, South West and Wales, excluding any purchases that were made in Northern Ireland due to small numbers (n<5).

### Ovarian cancer risk questionnaire (24 items)

Participants completed a short questionnaire about ovarian cancer risk factors including ethnicity, marital status, body mass index (BMI), age at menarche, menopausal status, age at menopause, parity, breastfeeding, hysterectomy, tubal ligation, cancer history, endometriosis, aspirin use, oral contraceptive use, hormone replacement therapy use, family history of ovarian and breast cancers, vaping and cigarette smoking. The questionnaire requested information about the symptoms experienced (if any) and number of visits to the general practitioner in the year leading up to participation for controls and cancer referral or diagnosis for cases. All responses were voluntary.

### Participant retention

Participant retention was measured at three stages. The first stage was the number of participants with a valid consent and completed ovarian cancer risk questionnaire. Participant retention was calculated based on withdrawals and valid consent (eg, provided consent for retailers).

The second stage assessed the number of control participants with valid identity verification details. Identity verification was recorded as ID verified or not ID verified.

The last stage of retention was assessed based on the confirmation of the records with the retailers following transactional data requests. If participants' reported card details were the same as the retailers' and data was received, this was recorded as complete participation. If their reported card details did not match the retailers' records, we recorded these participants as unmatched.

### Statistical analysis

Participation and data retention was recorded in a CONSORT diagram. A $\chi^2$ test was used to assess the differences between cases and controls based on the following characteristics: age (10-year age groups), ethnicity, number of people in the household, loyalty

card ownership and the location and IMD quintile of the stores where transactions took place. Each postcode for the stores where individuals purchased items were converted to the UK regions and IMD deciles using the English Indices of Deprivation 2019.[15] These were then recoded into quintiles ranging from most deprived to least deprived.[16] If the sample size was less than 5, groups were merged together to ensure safe data reporting. Missing data and questionnaire completion rates were calculated.

## RESULTS

### Recruitment timeline

The cumulative recruitment of CLOCS cases and controls is shown in figure 1. CLOCS recruitment began in December 2019, and at the beginning of the COVID-19 pandemic in the UK in March 2020, recruitment was paused to investigate the impact of the pandemic on recruitment while patients were not able to attend clinics.

Active control recruitment was also paused, however, opportunistic recruitment occurred through secondary sources. The changes were implemented, and patient recruitment resumed from September 2020. Facebook Advertising was used to recruit control participants from 17 September 2020 to 26 April 2021.

### Participant characteristics (included in ovarian cancer risk analysis)

The characteristics of CLOCS participants included in the final dataset for ovarian cancer risk analysis are shown in table 1.

### Cases

NHS recruitment sites reported that 306 patients were eligible and interested to take part in CLOCS. Of those, 183 patients (59%) returned their completed consent form and questionnaires. After exclusion due to insufficient consent, 182 were included in the final dataset for ovarian cancer risk analysis. About 70% of cases who

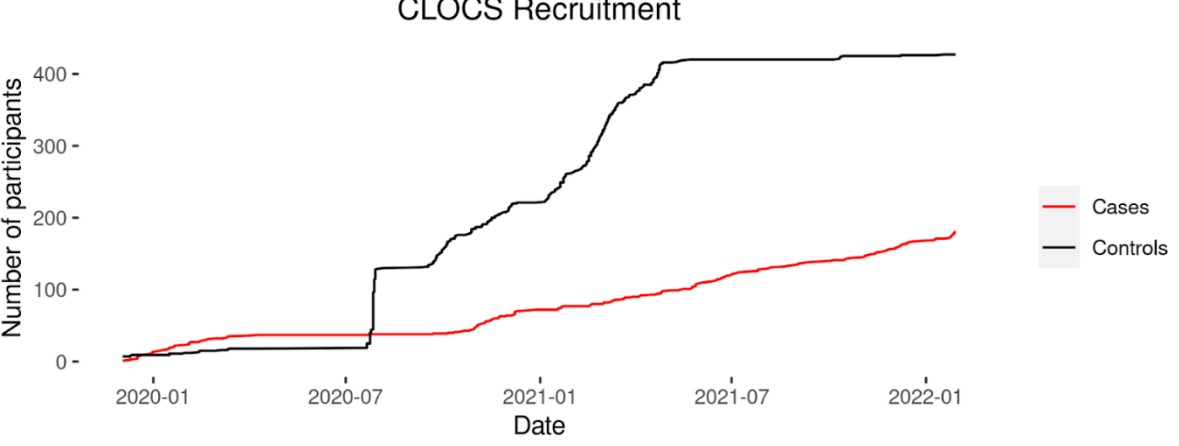

**Figure 1** CLOCS recruitment timeline for cases and controls. CLOCS, Cancer Loyalty Card Study.

**Table 1** Participant characteristics

| Characteristic | Cases (n=182) | | Controls (n=427) | | |
|---|---|---|---|---|---|
| | n | % | n | % | P value |
| Age | | | | | |
| 18–39 | 6 | 3.6 | 61 | 16.2 | <0.001 |
| 40–49 | 7 | 4.7 | 96 | 23.8 | |
| 50–59 | 32 | 21.3 | 125 | 31.8 | |
| 60–69 | 64 | 36.7 | 107 | 21.2 | |
| ≥70 | 73 | 33.7 | 33 | 5.9 | |
| Missing | 0 | 0.0 | 5 | 1.2 | |
| Race and ethnicity | | | | | |
| White | 173 | 95.1 | 400 | 93.7 | 0.392 |
| Non-white | 8 | 4.4 | 11 | 2.6 | |
| Prefer not to say | 0 | 0.0 | 0 | 0.0 | |
| Missing | 1 | 0.5 | 16 | 3.7 | |
| Number of people in the household | | | | | |
| 1 | 35 | 22.2 | 45 | 14.5 | <0.001 |
| 2 | 78 | 52.1 | 120 | 40.5 | |
| 3 | 17 | 11.8 | 62 | 20.9 | |
| 4 | 10 | 6.9 | 45 | 15.2 | |
| ≥5 | 2 | 1.4 | 24 | 8.1 | |
| Missing | 8 | 5.6 | 2 | 0.7 | |
| Family history of breast or ovarian cancer | | | | | |
| No | 139 | 76.4 | 327 | 76.6 | 1.00 |
| Yes | 43 | 23.6 | 99 | 23.2 | |
| Missing | 0 | 0.0 | 1 | 0.2 | |
| Loyalty card | | | | | |
| HSR1 only | 48 | 26.4 | 169 | 39.6 | 0.001 |
| HSR2 only | 44 | 24.2 | 116 | 27.2 | |
| Both HSR1 and HSR2 | 86 | 47.3 | 142 | 33.3 | |
| Neither | 4 | 2.2 | 0 | 0.0 | |
| Loyalty card use | | | | | |
| Not at all | 1 | 0.5 | 2 | 0.5 | 0.449 |
| Not very often | 9 | 4.9 | 10 | 2.3 | |
| Sometimes | 18 | 9.9 | 42 | 9.8 | |
| Often | 42 | 23.1 | 117 | 27.4 | |
| All the time | 109 | 59.9 | 256 | 60.0 | |
| Missing | 3 | 1.6 | 0 | 0.0 | |

HSR, high street retailer.

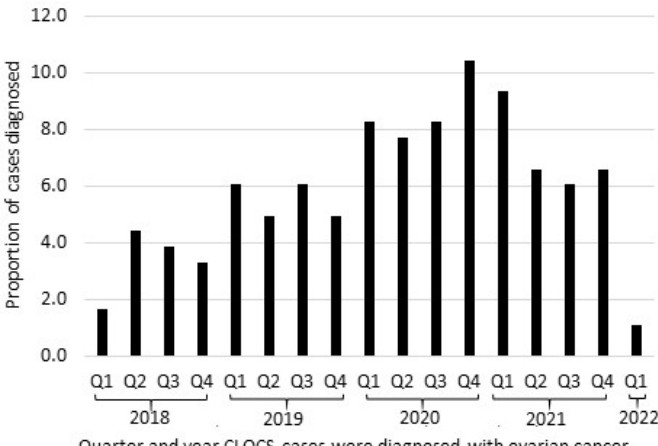

**Figure 2** Timeline of when CLOCS case participants were diagnosed with ovarian cancer by quarter (Q) and year. CLOCS, Cancer Loyalty Card Study.

living in a two-member household (52.1%), followed by 22.2% in a single-member household and 20.1% living in a household of three or more people. About 25% of cases had a family history of ovarian and/or breast cancer in first-degree, female relative.

### Controls
In total, there were 657 online submissions of the secure online questionnaire (see figure 2). Of those, 441 (61.7%) were considered valid control participants once duplicates and test submissions were removed. There were 14 participants who were excluded from the study; 10 withdrew from the study and 4 were ineligible. Therefore, 427 control participants were included in the final dataset for ovarian cancer risk analysis.

Compared with case participants, controls were younger with a larger proportion in age groups 40–49 years (23.8%) and 50–59 years (31.8%). There were also significantly fewer control participants recruited in the older age groups 60–69 years (21.2%) and 70 years and older (5.9%) compared with case participants (p<0.05). Similar to cases, most controls reported living in a two-member household (40.5%), 14.5% in a single-member household and 44.2% in a household with three or more individuals. Also, 76.6% of control participants reported having a first-degree, female relative with ovarian and/or breast cancer.

### Participant retention rates
The recruitment pathways for cases and controls and how participants were retained within the study are reported in the CONSORT diagram in figure 3.

While case participants did not require additional identity verification, there were four cases who did not provide loyalty card details, and therefore, 178 cases were eligible for data requests from the retailers. In total 134 (73.6%) case participants were eligible for data requests from the HSR1 and 130 (71.4%) case participants from HSR2. In

took part in CLOCS were diagnosed from January 2020 onward (figure 2).

Most cases were 60 years or older (70.4%). The majority of the cases identified themselves as white ethnic background (95.1%), and the rest of the cases were grouped together as non-white ethnic background due to small numbers (4.4%). Over half of case participants reported

## CLOCS CONSORT Flow Diagram

### Based on CLOCS participant data up to 31 Jan 2022

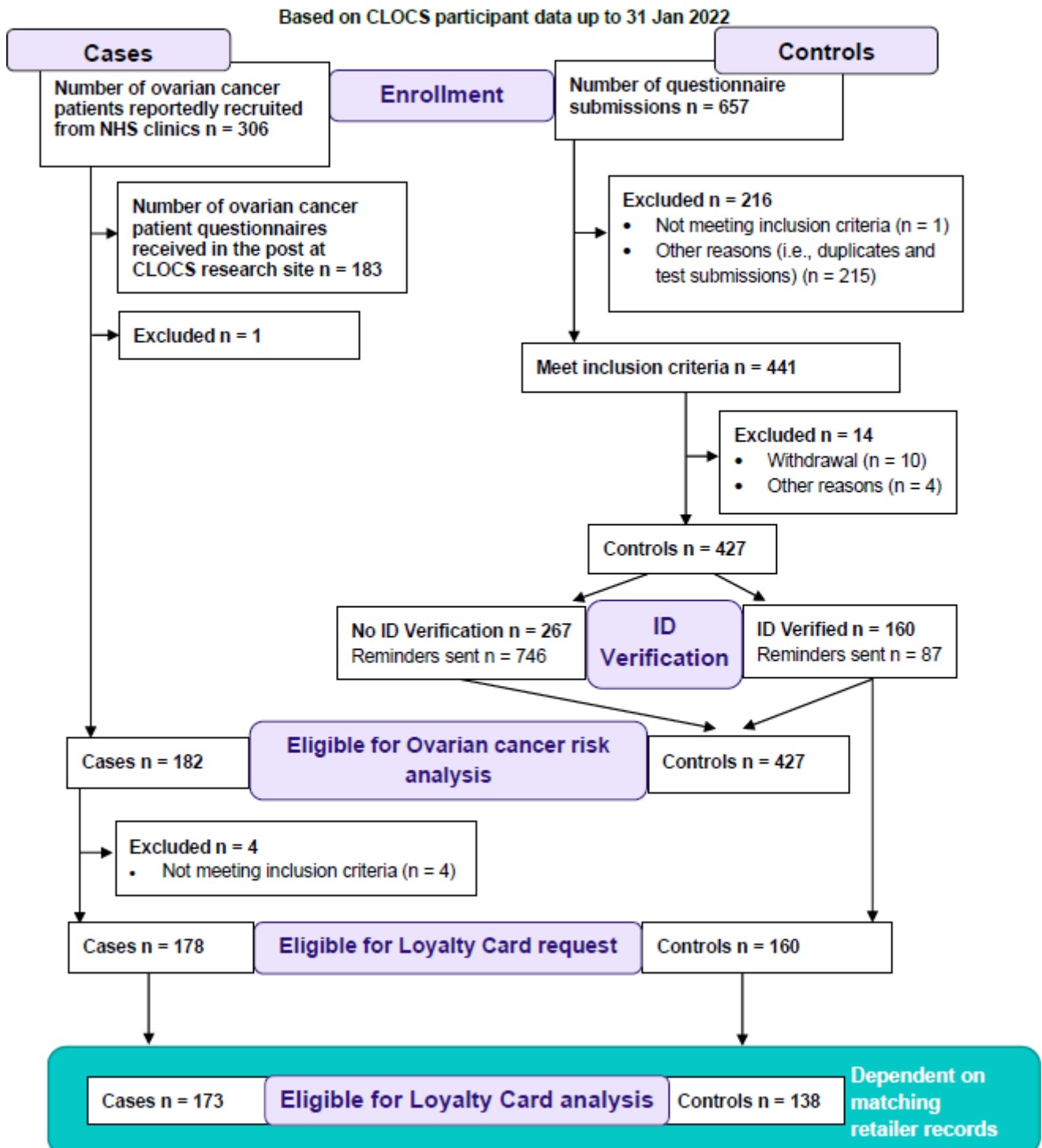

**Figure 3** CLOCS consort diagram detailing participant recruitment. CLOCS, Cancer Loyalty Card Study; ID, identification; NHS, National Health Service.

the end, transactional history data were received for 158 cases (88% case participant data retention rate) (table 2).

To maximise the number of ID verified control participants, over 700 email reminders were sent to participants asking them to complete the ID verification step. As of 31 January 2022, 160 (37.5%) control participants provided sufficient ID verification and were eligible for

purchase history requests on their behalf. At the end of the study, 137 control participant (85.6% control participant data retention rate) card details matched retailer records, allowing the research team to receive up to 6 years of their past transactional data (see table 3). Only 51% of control participants had records for both retailers. The association between age distribution and

**Table 2** Number of participants included in the transactional data analysis by age and retailers

| | HSR1 | | | | HSR2 | | | |
| | Cases (n=134) | | Eligible controls (n=120) | | Cases (n=130) | | Eligible controls (n=107) | |
| Age | Matched | % | Matched | % | Matched | % | Matched | % |
|---|---|---|---|---|---|---|---|---|
| 18–49 | 9 | 6.7 | 41 | 34.2 | 10 | 7.6 | 28 | 26.2 |
| 50–59 | 19 | 14.2 | 31 | 25.8 | 26 | 20.0 | 25 | 23.4 |
| 60–69 | 43 | 32.1 | 25 | 20.8 | 43 | 33.1 | 20 | 18.7 |
| 70+ | 58 | 43.3 | 9 | 7.5 | 48 | 36.9 | 10 | 9.3 |
| Total | 129 | 96.3 | 106 | 88.3 | 127 | 97.7 | 83 | 77.6 |

Eligible=ID verified.
HSR, high street retailer; ID, identification.

ID verification among control participants was not significant (p=0.80).

### Representativeness based on the store postcodes recorded on loyalty cards

Based on the transactions recorded on loyalty cards, we identified 697 postcodes for cases and 1127 postcodes for controls in nine UK regions, excluding Northern Ireland. The proportion of the distribution of the purchases carried out by cases and controls in each UK region were significantly different, p<0.001 (see table 3). More than five percentage points difference between cases and controls were recorded in transactions at East of England and Yorkshire and Humber (6.9% vs 18.5%), Midlands (4.3% vs 12.1%), South East (27.7% vs 17.6%) and Wales (15.1% vs 3.8%), respectively. The largest proportion of

**Table 3** The number of individual transaction postcodes recorded on loyalty cards among participants in each UK region

| | Transaction postcodes | | | |
| | Cases | | Controls | |
| UK region* | n† | % | n† | % |
|---|---|---|---|---|
| East of England and Yorkshire and Humber | 48 | 6.9 | 209 | 18.5 |
| London | 104 | 14.9 | 192 | 17.0 |
| North East | 24 | 3.4 | 36 | 3.2 |
| North West | 47 | 6.7 | 87 | 7.7 |
| Midlands | 30 | 4.3 | 136 | 12.1 |
| Scotland | 85 | 12.2 | 82 | 7.3 |
| South East | 193 | 27.7 | 198 | 17.6 |
| South West | 61 | 8.8 | 144 | 12.8 |
| Wales | 105 | 15.1 | 43 | 3.8 |
| Total | 697 | 100.0 | 1127 | 100.0 |

*Purchases at Northern Ireland were excluded due to small numbers (n<5).
†The transactions could have been recorded at one or more store postcodes.

purchases for both cases and controls were collected in South East.

However, there were no significant differences between IMD quintiles recorded for store locations where overall purchases took place between cases and controls, p=0.79 (see table 4). Among cases, 105 transactions (20.5%) took place in stores located in the most deprived quintile (1), comparison to 188 purchases (19.0%) among controls. The lowest numbers of transactions were recorded in stores located in the least deprived quintile for both cases (n=67, 13.1%) and controls (n=124, 12.5%).

### Cost per control participant recruitment

Most controls were recruited through paid social media advertisement (n=249, 58.3%), costing on average £15 per participant compared with opportunistic methods which were not paid (n=178, see table 5). About 31% (n=133) of control participants were recruited opportunistically during additional research that investigated attitudes towards CLOCS and willingness to take part, but their recruitment pathway was not recorded at the time (missing data, n=131, 31%). The cost per participant for those non-targeted studies was an average of £5.

**Table 4** The number of transactions recorded on loyalty cards among participants based on the Index of Multiple Deprivation (IMD) quintiles

| | Transactions | | | |
| | Cases | | Controls | |
| Index of Multiple Deprivation quintile | n* | % | n* | % |
|---|---|---|---|---|
| 1 (most deprived) | 105 | 20.5 | 188 | 19.0 |
| 2 | 129 | 23.5 | 261 | 26.4 |
| 3 | 121 | 23.7 | 231 | 23.4 |
| 4 | 98 | 19.2 | 185 | 18.7 |
| 5 (least deprived) | 65 | 13.1 | 124 | 12.5 |
| Total* | 511 | 100.0 | 989 | 100.0 |

*No postcode-IMD conversions were recorded for the store postcodes located in Wales and Scotland.

**Table 5** Source from which control participants reported they heard about the study and cost of recruitment per participant from each source

| Recruitment source | n | % | Cost per participant (GBP) |
|---|---|---|---|
| From a patient | 4 | 0.9 | 0.00 |
| HSR email | 4 | 0.9 | 0.00 |
| Twitter | 2 | 0.5 | 0.00 |
| Facebook | 249 | 58.3 | 15.00 |
| Instagram | 3 | 0.7 | 0.00 |
| Press | 1 | 0.2 | 0.00 |
| Word of mouth | 23 | 5.4 | 0.00 |
| VOICE | 8 | 1.9 | 0.00 |
| Missing* | 133 | 31.1 | 5.00 |
| Total | 427 | 100.0 | |

*Number of participants recruited prior to amendment to the research protocol in September 2020 that included a new item measuring the source of recruitment.

### Risk factor questionnaire completion rates

The majority of the items on the risk factor questionnaire were completed by both cases and controls, with the largest proportion of missing data observed for menopausal status among control participants (21.08%), followed by breastfeeding duration (18.50%) and BMI (6.79%). Both cases and controls had similar missing values for the number of full-term birth (4.40% vs 6.09%), the number of non-full-term birth (6.59% vs 7.26%) and age at first birth (3.85% vs 4.22%). The distribution of the missing data is reported in online supplemental table 1.

### DISCUSSION

The participant recruitment and data retention in this observational case–control study demonstrate that this is a feasible method and is acceptable to the public to investigate individual transactional data for health research. This is a major achievement in social sciences and epidemiological research to further understand the impact of individual behaviours on health outcomes with robust and objective datasets. Furthermore, despite the challenges of the COVID-19 pandemic, when recruitment was finalised on 31 January 2022, the study was only 73 participants short of the initial target to recruit 500 control participants. CLOCS recruited approximately 10 cases and 25 controls per month over the course of 17 active recruitment months. Most control participants were willing to take part (n=427) and provided their consent for the researchers to request their transactional loyalty card history from the high street retailers. However, the researchers were unable to process over two-thirds of the control participants consent due to the lack of identity verification details. Furthermore, about 25% and 14% of cases and controls who provided information, including the name on the card and the long digit card number,

did not match with the retailers' records, resulting in only 173 cases and 138 controls included in the final transactional data analyses.

The representativeness of the study population was assessed with the comparison of the population characteristics as well as the comparison of the population based on the postcode level characteristics of the store locations recorded on the loyalty cards. A key outcome is younger individuals without ovarian cancer were more willing to take part than those who were 60 years and older, despite all age groups being targeted as part of the recruitment process. This sampling bias is commonly observed in many survey studies with broad inclusion criteria for those who are eligible to take part[12] and unable to reach the sample size without careful targeting. On the other hand, CLOCS case participants were representative of the age distribution of patients with ovarian cancer in the UK[3] and were recruited across 12 research sites. The imbalance in age between cases and controls is a potential barrier for 1:1 case and control age-matched analyses as the majority of patients with ovarian cancer were aged 60 years old and above. Therefore, it is proposed that 2:1 age matching with replacement is used adjusting for number of people in the household, and ovarian cancer risk factors that are found to be significantly associated with ovarian cancer risk among CLOCS participants. This solution to case–control matching is in line with other case–control studies with insufficient control participants to match each case.[17]

It is important to note that the High Street Retailers were not significantly involved in recruitment to CLOCS. A pilot study of an email from one High Street Retailer resulted in four control participants (table 5). The remaining control participants (n=423) were recruited directly by the researchers through various sources. However, the limitation is acknowledged that control participants who have chosen to participate might be biased due to frequent loyalty card use or more health conscious than the patients with cancer which may be reflected in their shopping habits. There were also some contextual barriers as an outcome of the COVID-19 pandemic to be considered for future research. It is suggested that the way the UK government collected data to inform public health strategies may have had a negative influence on people's willingness to take part in studies like CLOCS. In July 2020, after the first lockdown restrictions were eased in the UK,[18] we carried out a population-based survey in England asking individuals their willingness to share commercial data including shopping, internet searches, wearable devices and social media data for health research.[19] A key finding of this study was that two-thirds of the population were willing to share their commercial data with academics for health research, and willingness to share data reduced with the participants' ages. Also, only half of the respondents were happy to share shopping data (51.8 %) although it was more acceptable compared with internet searches (35.2%), smart phone applications (32%), wearable

devices (31.8%) and social media data (30.5%).[19] These findings are reflective of the reduction in public trust in the government and the negative perceptions towards the contact tracing app that collected personal information using mobile phone data.[20 21]

Under Article 15 in the General Data Protection Regulation (GDPR) in the UK, individuals have the right to access and request a copy of their data, often referred to as a subject access request.[22] There is growing literature on impersonating to take part in research for financial gains and/or trying to get access to sensitive information.[23] Transactional data are considered sensitive data and are, therefore, subject to strict data security measures by data controllers. Although ID verification is not required by all organisations, it is recommended for researchers to verify the authenticity of the participants so that data of individuals are not shared with third party organisations unlawfully. It is possible that the process of requesting ID verification details in CLOCS may not have been as clear to the participants who agreed to take part in the study resulting in dropouts. Previous studies have shown that individuals only read part of the study information and the consent form[24 25] which may explain why 14 control participants withdrew after they were asked to provide ID verification details and a majority did not respond to our email reminders. A potential solution to reduce the number of dropouts could be to use dynamic consent processes and two-step authentication processes that are commonly used in other secure access scenarios such as online banking and shopping.[26] Dynamic consent gives the user more control over when they want to share their data, how they want to share it, and allows researchers to amend the consent if necessary to add new variables and datasets.[26]

In conclusion, this paper demonstrates the feasibility outcomes of CLOCS and discusses how to resolve the issues in the future to maximise participation and data retention rates. Specifically, it shows that although the general public is amenable to taking part and providing consent for their data to be shared with researchers, due to robust application of the data sharing principles based on GDPR and Data Protection Act 2018 by the researchers and the high street retailers, only a third of the dataset was included for the control participants. The contextual and procedural barriers as well as the public acceptability of data sharing should be considered in the design of future studies to ensure that individuals' contributions to health research with transactional data are successful.

**Author affiliations**
[1]Department of Surgery and Cancer, Imperial College London, London, UK
[2]MRC Centre for Environment and Health, Imperial College London, London, UK
[3]Epidemiology and Biostatistics, School of Public Health, Imperial College London, London, UK
[4]School of Public Health, Imperial College London, London, UK
[5]Cancer Sciences, University of Birmingham, Birmingham, UK
[6]Epidemiology and Public Health, University College London, London, UK

**Acknowledgements** The authors would like to acknowledge all of the participants in CLOCS, the participating retailers, and the NHS sites that recruited patients with ovarian cancer to this study. We would like to acknowledge Deborah Tanner and Fiona Murphy who have acted as ovarian cancer patient representatives since the inception of the CLOCS project.

**Contributors** JF, YH and HRB conceived the study. HRB and YH conducted all statistical analyses. HRB, JF and YH reviewed the results. EJ provided expertise in information governance and data management. HRB wrote the first draft of the manuscript. YH, JF, SS and MC-H commented and reviewed the final draft of the manuscript. JF is responsible for the overall content as the guarantor. All authors agree on the final version of the manuscript.

**Funding** This study was funded by Cancer Research UK Early Detection and Diagnosis project grant (C38463/A26726) with generous support from the Peter Sowerby Foundation and additional support from the Cancer Research UK Imperial Centre, the National Institute for Health Research Imperial Biomedical Research Centre and the Ovarian Cancer Action Research Centre. YH and HRB were funded by CRUK.

**Competing interests** None declared.

**Patient and public involvement** Patients and/or the public were involved in the design, or conduct, or reporting, or dissemination plans of this research. Refer to the Methods section for further details.

**Patient consent for publication** Not applicable.

**Ethics approval** This study involves human participants and was approved by North West-Greater Manchester South Research Ethics Committee Reference (19/NW/0427). Participants gave informed consent to participate in the study before taking part.

**Provenance and peer review** Not commissioned; externally peer reviewed.

**Data availability statement** Data are available upon reasonable request. Access to all CLOCS purchase history data collected from the retailers is restricted due to the data sharing agreements in place and the sensitive nature of the data. We, however, welcome other researchers to contact us for collaborations. The PI and research team will review each proposal and, if approved, the analysis will be conducted by a member of the CLOCS team.

**ORCID iDs**
Hannah R Brewer http://orcid.org/0000-0001-5495-8597
James Flanagan http://orcid.org/0000-0003-4955-1383
Yasemin Hirst http://orcid.org/0000-0002-0167-9428

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
