## [Reviewer comments · BMJ Open]

ARTICLE DETAILS

TITLE (PROVISIONAL)	Cancer Loyalty Card Study (CLOCS): feasibility outcomes for an observational case-control study focusing on the patient interval in ovarian cancer
AUTHORS	Brewer, Hannah; Chadeau-Hyam, Marc; Johnson, Eric; Sundar, Sudha; Flanagan, James; Hirst, Yasemin

VERSION 1 – REVIEW

REVIEWER	LIAQAT HUSSAIN Government College University Faisalabad
REVIEW RETURNED	26-Oct-2022

GENERAL COMMENTS	It is good written manuscript, but I have few concerns; Needed more clarity in study objectives and outcomes and authors are suggested to please be more specific. Discussion should also needed revision with more recent citations.
---

REVIEWER	Martin Stockler University of Sydney , NHMRC Clinical Trials Centre
REVIEW RETURNED	18-Nov-2022

GENERAL COMMENTS	I enjoyed reading this well-conceived, -conducted, -analysed, and -written study evaluating the feasibility of obtaining transactional data from loyalty cards to perform a cancer case-control study. As reported by the authors, the study confirms that this approach is feasible, but has challenges. The main value of this paper is disseminating the method and the lessons learnt from trying to implement it. My only suggestion for improvement is to report p-values only for the statistical hypothesis tests, and omit the chi-squared values and degrees of freedom because they contribute no additional useful information.
--

REVIEWER	Katherine Lawson-Michod University of Utah Health Huntsman Cancer Institute
REVIEW RETURNED	26-Jan-2023

GENERAL COMMENTS	Overview of the paper In this study, the author's goal was to evaluate the feasibility of a UK-based case-control study using loyalty cards as an opportunity for ovarian cancer symptom surveillance. Cases were identified using their NHS number, and controls were identified using social media campaigns, word of mouth, snowball methods, and targeted email campaigns by the retail shops accepting the loyalty cards. The
--

	authors present data on response rates and participant characteristics. Overall, this was an interesting manuscript which relates to an important topic, the poor survival for ovarian cancer driven largely by the late stage at diagnosis. In addition, the manuscript offers important data and commentary regarding research participation for both cases and controls in research leveraging transactional data. Major comments  - My major comments revolve around the study's overall impact and the manuscript's readability. Regarding impact, there appeared to be inconsistencies in the study's overall goal. For example, in paragraph two of the introduction, it is stated that the rationale for this study is to test the hypothesis of identifying an increase in medication use among ovarian cancer patients using loyalty card data. However, later, it is stated that the paper aims to present the feasibility outcomes of the study, including participation rates and participant characteristics, to optimize data collection in the future. The discussion appears to focus on the second aim. I suggest clarifying the overall aim of the study throughout the manuscript. - The authors also discussed two separate analytic sets, the ovarian cancer risk set and the loyalty card analysis set. However, the authors did not state the inclusion/exclusion for the ovarian cancer risk set or discuss the feasibility results for this set. Please provide further clarification throughout the manuscript regarding the purpose of these two analytic sets. General  - Some acronyms are not defined (e.g., NHS). - Inconsistencies in formatting (e.g., use of November vs. Jan, use of sentence case, and table formatting). Abstract  - "Identification of any barriers to recruitment" appears to belong in the discussion rather than the results section. Introduction  - Some statements lack sufficient citations, including: "While there has been progress in efforts to reduce diagnostic delays using routine blood tests". The introduction starts by discussing the feasibility study in the context of ovarian cancer. This section might benefit from discussing major barriers to earlier diagnosis of ovarian cancer, including the ineffectiveness of population-level screening to date and nonspecific/vague symptoms. Methods  - Please discussion eligibility criteria for the ovarian cancer risk analysis set. - In-person approach before COVID-19 is suggested but not explicitly stated in this section. Consider updating wording to improve clarity. - In the second to last sentence, it is unclear which forms were completed by the clinical care team. "If they decided to take part, they were given the consent form, a risk factor questionnaire, and a clinical questionnaire, which was completed by their clinical care team, to either complete in the clinic or take with them to complete at another time." Please update wording to improve clarity. - Description of informed consent could be simplified to improve readability. - Rationale for ID verification (Starting with "Under Article 15 ...") could also be shorted to improve readability. Specifically, suggest moving information about Article 15 to the discussion section. - Please state whether any patients were recontacted. - Race and ethnicity appear to be conflated in the participant
--	--

	descriptive characteristics section. Please update “ethnicity” to “race and ethnicity” as white refers to a racial group. Results  - Under Cases the author’s state: “Of those, 183 patients (59%) returned their completed consent form and questionnaires. After exclusion, 182 were included in the final dataset for ovarian cancer risk analysis.”. However, reason for exclusion is not stated here or elsewhere. Please state reason for exclusion. - Similar comment as above--under Controls please state the reason for exclusion and ineligibility. Discussion The author's discussion of the limitations of their study only addresses recruitment during the COVID-19 pandemic and differences in the age distribution for cases and controls. The authors propose using transactional data for cases and controls to evaluate differences in purchasing patterns for cases compared to controls. As discussed by the authors, the premise for this research is to use transactional history on loyalty cards for cancer symptom surveillance for ovarian cancer. However, the authors did not discuss that control selection is likely dependent on the exposure (i.e., loyalty card transactions). While cases were identified through NHS clinics following ovarian cancer diagnosis and excluded if they did not have a loyalty card, controls were recruited through targeted marketing by the high street retailers, which offer loyalty cards and other social media campaigns. Individuals who are likely to respond to targeted marketing campaigns and advertisements from HSR promoting control recruitment are likely to be more frequent users of the loyalty card program. Indeed, in table 3, controls had a higher incidence of transactions than cases across almost all geographic regions and this difference was statistically significant. Please consider this in your discussion of limitations. Tables  - Overall, these were really clear and presented pertinent results, however there is some inconsistent formatting including cell alignment and sentence case.
--	--

VERSION 1 – AUTHOR RESPONSE

1. Reviewer: 1

Dr. LIAQAT HUSSAIN, Government College University Faisalabad Comments to the Author:
It is good written manuscript, but I have few concerns; Needed more clarity in study objectives and outcomes and authors are suggested to please be more specific.
Discussion should also needed revision with more recent citations.

We have now updated the introduction to make it clear what the objectives of this manuscript were, in contrast to the objectives of the whole CLOCS project. We have also added additional recent citations to the discussion.

2. Reviewer: 2

Prof. Martin Stockler, University of Sydney Comments to the Author:
I enjoyed reading this well-conceived, -conducted, -analysed, and -written study evaluating the feasibility of obtaining transactional data from loyalty cards to perform a cancer case-control study.

As reported by the authors, the study confirms that this approach is feasible, but has challenges. The main value of this paper is disseminating the method and the lessons learnt from trying to implement it.

My only suggestion for improvement is to report p-values only for the statistical hypothesis tests, and omit the chi-squared values and degrees of freedom because they contribute no additional useful information.

Chi- square values/degrees of freedom have been removed from the text.

6. Major comments

- **My major comments revolve around the study's overall impact and the manuscript's readability. Regarding impact, there appeared to be inconsistencies in the study's overall goal. For example, in paragraph two of the introduction, it is stated that the rationale for this study is to test the hypothesis of identifying an increase in medication use among ovarian cancer patients using loyalty card data. However, later, it is stated that the paper aims to present the feasibility outcomes of the study, including participation rates and participant characteristics, to optimize data collection in the future. The discussion appears to focus on the second aim. I suggest clarifying the overall aim of the study throughout the manuscript.**

The Cancer Loyalty Card Study (CLOCS) protocol and main results have been published elsewhere: <https://bmjopen.bmj.com/content/10/9/e037459.info>

<https://publichealth.jmir.org/2023/1/e41762/>

We re-state the aims of CLOCS in the introduction to provide context for the purpose of this paper which is to present the feasibility outcomes. We have now noted that the main CLOCS results have been published with the corresponding citation.

3. - **The authors also discussed two separate analytic sets, the ovarian cancer risk set and the loyalty card analysis set. However, the authors did not state the inclusion/exclusion for the ovarian cancer risk set or discuss the feasibility results for this set. Please provide further clarification throughout the manuscript regarding the purpose of these two analytic sets.**

We have included the inclusion/exclusion criteria for enrolling in the study, which is the criteria for the cancer risk dataset (including all participants). The additional criteria for being included in the loyalty card analysis was providing ID verification documents and matching name and card numbers to allow the data request from the retailers.

8. General

- **Some acronyms are not defined (e.g., NHS).**

Acronyms have now been defined where needed.

9. - **Inconsistencies in formatting (e.g., use of November vs. Jan, use of sentence case, and table formatting).**

Date formats have been amended and are now consistent.

10. Abstract

- **"Identification of any barriers to recruitment" appears to belong in the discussion rather than the results section.**

This was in the outcomes section of the abstract, not the results, and this is appropriate given that it is one of the outcomes we were aiming to assess.

11. Introduction

- **Some statements lack sufficient citations, including: "While there has been progress in efforts to reduce diagnostic delays using routine blood tests".**

This statement has one reference [3] and we have added two additional references to support this claim.

[4] Barr CE, Funston G, Jeevan D, Sundar S, Mounce LTA, Crosbie EJ. The Performance of HE4 Alone and in Combination with CA125 for the Detection of Ovarian Cancer in an Enriched Primary Care Population. *Cancers (Basel)*. 2022 Apr 24;14(9):2124. doi: 10.3390/cancers14092124. PMID: 35565253; PMCID: PMC9101616.

[5] Davenport C, Rai N, Sharma P, Deeks JJ, Berhane S, Mallett S, Saha P, Champaneria R, Bayliss SE, Snell KI, Sundar S. Menopausal status, ultrasound and biomarker tests in combination for the diagnosis of ovarian cancer in symptomatic women. *Cochrane Database Syst Rev*. 2022 Jul 26;7(7):CD011964. doi: 10.1002/14651858.CD011964.pub2. PMID: 35879201; PMCID: PMC9314189.

12. The introduction starts by discussing the feasibility study in the context of ovarian cancer. This section might benefit from discussing major barriers to earlier diagnosis of ovarian cancer, including the ineffectiveness of population-level screening to date and nonspecific/vague symptoms.

We have expanded the section in the introduction with the following text (added text in Blue):

13. Late-stage ovarian cancer diagnosis is often associated with delayed patient presentation and a longer diagnostic interval due to non-specific cancer symptoms (e.g., feeling bloated, indigestion, feeling full, abdominal pain). While there has been progress in efforts to reduce diagnostic delays using routine blood tests (i.e., CA125) [3] there is little evidence focusing on women's self-care behaviours associated with managing non-specific symptoms. It has been demonstrated that population screening of asymptomatic women is not effective in reducing mortality from the disease and is, therefore, not recommended in the UK [ref]. Screening in the symptomatic population may prove more effective [ref], however, the main barrier to earlier diagnosis remains a lack of awareness of the vague non-specific symptoms of ovarian cancer [ref].

14. Methods

- **Please discuss eligibility criteria for the ovarian cancer risk analysis set.**

The eligibility criteria for the ovarian cancer risk analysis set were the same for the loyalty card analysis set except for the need for matching loyalty card details and ID verification (for controls). We have added text to clarify this.

15. - In-person approach before COVID-19 is suggested but not explicitly stated in this section. Consider updating wording to improve clarity.

This has been added.

16. - In the second to last sentence, it is unclear which forms were completed by the clinical care team. "If they decided to take part, they were given the consent form, a risk factor questionnaire, and a clinical questionnaire, which was completed by their clinical care team, to either complete in the clinic or take with them to complete at another time." Please update wording to improve clarity.

Changed punctuation so that 'completed by their clinical care team' is associated with the clinical questionnaire. "If they decided to take part, participants were given the consent form, a risk factor questionnaire, and a clinical questionnaire (completed by their clinical care team) to either complete in the clinic or take with them to complete at another time."

17. - Description of informed consent could be simplified to improve readability.

We have not made changes to this text describing the informed consent as this is an important topic for future research projects, and were unable to reduce it without losing meaning.

18. - Rationale for ID verification (Starting with "Under Article 15 ...") could also be shorted to improve readability. Specifically, suggest moving information about Article 15 to the discussion section.

We have now made this change by moving some of this text to the discussion.

19. - Please state whether any patients were recontacted.

39 patients were recontacted to verify loyalty card details, however, no participants were recontacted to participate in additional studies. We have added this text to the methods section.

20. - Race and ethnicity appear to be conflated in the participant descriptive characteristics section. Please update "ethnicity" to "race and ethnicity" as white refers to a racial group.

This has been updated.

21. Results

- Under Cases the author's state: "Of those, 183 patients (59%) returned their completed consent form and questionnaires. After exclusion, 182 were included in the final dataset for ovarian cancer risk analysis.". However, reason for exclusion is not stated here or elsewhere. Please state reason for exclusion.

This has been amended in the Cases section.

22. - Similar comment as above--under Controls please state the reason for exclusion and ineligibility.

The Controls section already includes the reasons for exclusion: duplicates, test submissions, withdrawals, and ineligibility.

23. Discussion

The author's discussion of the limitations of their study only addresses recruitment during the COVID-19 pandemic and differences in the age distribution for cases and controls. The authors propose using transactional data for cases and controls to evaluate differences in purchasing patterns for cases compared to controls. As discussed by the authors, the premise for this research is to use transactional history on loyalty cards for cancer symptom surveillance for ovarian cancer. However, the authors did not discuss that control selection is likely dependent on the exposure (i.e., loyalty card transactions). While cases were identified through NHS clinics following ovarian cancer diagnosis and excluded if they did not have a loyalty card, controls were recruited through targeted marketing by the high street retailers, which offer loyalty cards and other social media campaigns. Individuals who are likely to respond to targeted marketing campaigns and advertisements from HSR promoting control recruitment are likely to be more frequent users of the loyalty card program. Indeed, in table 3, controls had a higher incidence of transactions than cases across almost all geographic regions and this difference was statistically significant. Please consider this in your discussion of limitations.

It is important to note that the High Street Retailers were not significantly involved in recruitment to CLOCS. A pilot study of an email from one High Street Retailer resulted in 4 new participants (Table 5). The remaining control participants (n=423) were recruited directly by the researchers through various sources. However, the limitation is acknowledged that control participants who have chosen to participate might be biased due to frequent loyalty card use or health conscious.

We have added this text to the discussion.

24. Tables

- Overall, these were really clear and presented pertinent results, however there is some inconsistent formatting including cell alignment and sentence case.

Table and caption formatting have been updated to be consistent.

VERSION 2 – REVIEW

REVIEWER	Katherine Lawson-Michod University of Utah Health Huntsman Cancer Institute
REVIEW RETURNED	24-Mar-2023

GENERAL COMMENTS	The authors have addressed all comments in the review and I have no further comments.
---